# Gene-regulation modules in nonalcoholic fatty liver disease revealed by single-nucleus ATAC-seq

Fumihiko Takeuchi[1,2,3] , Yi-Qiang Liang[1], Hana Shimizu-Furusawa[4] , Masato Isono[1], Mia Yang Ang[1,5], Kotaro Mori[2] , Taizo Mori[6], Eiji Kakazu[6], Sachiyo Yoshio[6], Norihiro Kato[1,2,5]

We investigated the progression of nonalcoholic fatty liver disease from fatty liver to steatohepatitis using single-nucleus and bulk ATAC-seq on the livers of rats fed a high-fat diet (HFD). Rats fed HFD for 4 wk developed fatty liver, and those fed HFD for 8 wk further progressed to steatohepatitis. We observed an increase in the proportion of inflammatory macrophages, consistent with the pathological progression. Utilizing machine learning, we divided global gene regulation into modules, wherein transcription factors within a module could regulate genes within the same module, reaffirming known regulatory relationships between transcription factors and biological processes. We identified core genes—central to co-expression and protein–protein interaction—for the biological processes discovered. Notably, a large part of the core genes overlapped with genes previously implicated in nonalcoholic fatty liver disease. Single-nucleus ATAC-seq, combined with data-driven statistical analysis, offers insight into in vivo global gene regulation as a combination of modules and assists in identifying core genes of relevant biological processes.

## Introduction

Nonalcoholic fatty liver disease (NAFLD) is a continuum of liver abnormalities ranging from NAFL to nonalcoholic steatohepatitis, with a global prevalence of ~25% (Friedman et al, 2018). NAFLD can gradually develop from fatty liver to steatohepatitis, cirrhosis, and eventually, hepatic cancer. There are currently no approved drugs for NAFLD, and existing biomarkers are inadequate. The liver is composed of multiple cell types, including hepatocytes, cholangiocytes, stellate cells, endothelial cells, and white blood cells, each with a distinct role in disease development. Recent single-cell RNA-seq studies have revealed cell-type-specific gene expression changes in NAFLD (Ramachandran et al, 2019; Xiong et al, 2019; Chu et al, 2020).

A key upstream event in gene expression regulation is the binding of transcription factors (TFs) to DNA. ChIP-seq/Cut&Run/Cut&Tag assays can be performed on single cells, but they can only measure the binding of a few types of TF at a time (Salma et al, 2023). In contrast, single-nucleus ATAC-seq (snATAC-seq) detects open chromatin sites, where TFs with DNA-binding motifs that match the site sequence are presumed to bind, thereby providing information on the binding of all types of TFs simultaneously. Using snATAC-seq data, recent algorithms have inferred gene regulation by TFs, forming a gene regulatory network (GRN) for individual cell types (Fleck et al, 2022; González-Blas et al, 2022 Preprint; Jiang et al, 2022; Kamal et al, 2022 Preprint; Kartha et al, 2022; Wang et al, 2022 Preprint; Kamimoto et al, 2023). Unlike previous studies which focused on identifying the wiring of the GRN, our aim is to partition the entire GRN and extract major components that correspond to biological processes occurring in the relevant cells, such as lipid metabolism or inflammation in NAFLD. The binding of TFs to chromatin could not be analyzed in previous single-cell analyses of NAFLD, as they were all based on RNA-seq and not ATAC-seq.

The cell-type composition of tissues can change during disease development in various diseases and will impact the outcome. Although cell type counting of solid tissue has become feasible through single-cell assays, these assays are still expensive. Instead, deconvolution, which is the statistical inference of the cell type composition of tissue samples assayed in bulk, has been used for transcriptome and DNA methylation assays (Teschendorff & Zheng, 2017; Sturm et al, 2019). For ATAC-seq, Li and associates (Li et al, 2020) previously proposed an algorithm for deconvolution, but no experimental validation with single-cell data has been conducted.

Aiming to understand the cell-type composition and global TF regulation in NAFLD development, we performed single-nucleus and bulk ATAC-seq on the liver of rats fed or withdrawn from a high-fat diet (HFD). We first analyzed the cell-type composition in the liver and the gene expression separately by cell type. Then, to

[1]Department of Gene Diagnostics and Therapeutics, Research Institute, National Center for Global Health and Medicine, Tokyo, Japan [2]Medical Genomics Center, Research Institute, National Center for Global Health and Medicine, Tokyo, Japan [3]Systems Genomics Laboratory, Baker Heart and Diabetes Institute, Melbourne, Australia [4]Department of Hygiene and Public Health, School of Medicine, Teikyo University, Tokyo, Japan [5]Department of Clinical Genome Informatics, Graduate School of Medicine, The University of Tokyo, Tokyo, Japan [6]Department of Liver Diseases, The Research Center for Hepatitis and Immunology, National Center for Global Health and Medicine, Chiba, Japan

Correspondence: fumihiko@takeuchi.name

investigate global TF regulation in each cell type, we developed a novel statistical method that decomposes TF regulation into major modules, each being characterized by biological processes. We also developed a statistical method to detect core genes for the individual biological processes. Taking snATAC-seq data from an in vivo experimental model as input, our data-driven reductionist workflow proceeds from global TF regulation to TF regulation modules, then to biological processes, and finally to core genes. The core genes thus identified significantly overlap with previously known NAFLD genes, thereby providing experimental validation for our newly developed statistical methods.

## Results

### Cell type composition of the liver

Spontaneously hypertensive rats (SHR) fed a HFD for 4 wk developed fatty liver, and those maintained on HFD for 8 wk further developed inflammation and premature fibrosis (steatohepatitis) (Fig 1A and Table S1). Under the washout condition, where 4 wk of HFD is followed by 4 wk of a normal diet, fatty liver was partially ameliorated. Along with control rats fed only a normal diet, five dietary conditions were examined.

To catalog cell types in the liver, we performed snATAC-seq on one animal per dietary condition, totaling five animals. We obtained a total of 14,486 nuclei (~3,000 per sample), which were grouped into 16 clusters based on the similarity of chromatin opening. Hepatocytes (seven clusters), endothelial cells (three clusters), stellate cells (two clusters), and white blood cells (four clusters) were clearly separated (Fig 1B and C). The clusters for white blood cells corresponded to inflammatory macrophages, noninflammatory macrophages, T cells and NK cells combined, and B cells. Cholangiocytes were included in the hepatocyte clusters, whereas neutrophils and Kupffer cells were grouped into the macrophage clusters.

To measure the changes in cell-type population through dietary intervention, we performed bulk ATAC-seq on four animals per condition, totaling 20 animals. Using snATAC-seq as reference, we inferred the cell-type composition in the bulk ATAC-seq samples by deconvolution. Beforehand, using the five samples assayed in both single-nucleus and bulk ATAC-seq, the accuracy of deconvolution for ATAC-seq was validated (Fig 1D). The difference between the inferred proportion and the true proportion counted by snATAC-seq had a mean of zero and a SD of 0.035, indicating high accuracy. Considering that we used four animals per condition, the SE was calculated as 0.018. Compared with the normal diet condition, hepatocytes significantly decreased after 8 wk of HFD (Fig 1E). Inflammatory macrophages largely increased after 4 wk of HFD and further increased after 8 wk of HFD. Noninflammatory macrophages also increased after 8 wk of HFD. In addition, B cells increased under HFD conditions. Consistent with our findings, previous studies on HFD-induced NAFLD animal models reported an increase in liver-resident macrophages (Kiki et al, 2007), recruitment of monocyte-derived macrophages (Zhong et al, 2019), and an increase in intrahepatic B cells (Zhang et al, 2016; Barrow et al, 2021).

Under the washout condition, the cell-type composition reverted to the same as the normal diet condition. Having elucidated how the quantity of cell types changed through dietary intervention, we next investigate the qualitative change.

### Differential gene expression under HFD in each cell type

We analyzed cell-type-specific differential gene expression by comparing nuclei of the same cell type across different dietary conditions (Table S2 and Supplemental Data 1). Hereafter, clusters for inflammatory and noninflammatory macrophages are combined and referred to as the macrophage cell type. T cells, NK cells, and B cells were not included in the analysis because of the limited number of nuclei. The expression level of a gene in a nucleus was quantified as a score based on chromatin opening in the gene body and its vicinity. The biological processes characteristic of the differentially expressed genes were cell-type-dependent (Fig 2). Steroid metabolism and fatty acid metabolism genes were expressed in hepatocytes differently under HFD and washout compared with the normal diet. As for inflammation genes, their expression scores altered in macrophages after 4 wk of HFD and washout. Inflammation genes and apoptosis genes were expressed in hepatocytes differently after 8 wk of HFD. Genes related to the actin filament-based process were expressed in stellate cells differently after 8 wk of HFD.

Conventional differential gene expression analysis using bulk tissue samples has limitations, including the difficulty of detecting (1) changes in minor cell types, (2) changes in a cell type with low basal gene expression relative to other cell types, and (3) concurrent changes in gene expression and cell population that mutually cancel out, for example, down-regulated genes in a cell type that is increasing in population. We cross-checked cell-type-specific differential gene expression with bulk tissue transcriptome analysis for the comparison between the 4 wk of HFD and the normal diet. We found the results to replicate under the above-mentioned limitations (Fig S1). Differential gene expression was replicated in hepatocytes, but to a lesser degree in minor cell types. For macrophages, whose proportion increased after 4 wk of HFD, the up-regulated genes were replicated in bulk experiments but the down-regulated genes were not.

### Major modules of TF regulation and associated biological processes

To elucidate the global picture of TF regulation, we inferred the regulatory impact of all TFs on all genes and extracted major modules from these regulatory relationships. The genome-wide binding of a TF in a nucleus was quantified as a score based on the enrichment of the TF's motif in the open chromatin. Utilizing snATAC-seq, we quantified in all nuclei of each cell type (from rats under all dietary conditions) the genome-wide binding of TFs and the expression of genes, and inferred the regulation of the latter by the former by applying machine learning. The inferred impact of a TF on a gene was represented as a fractional number, corresponding to the weight of a GRN edge connecting the TF to the gene.

The regulatory relationships were subsequently divided into modules. A module comprises a subset of TFs and a subset of genes,

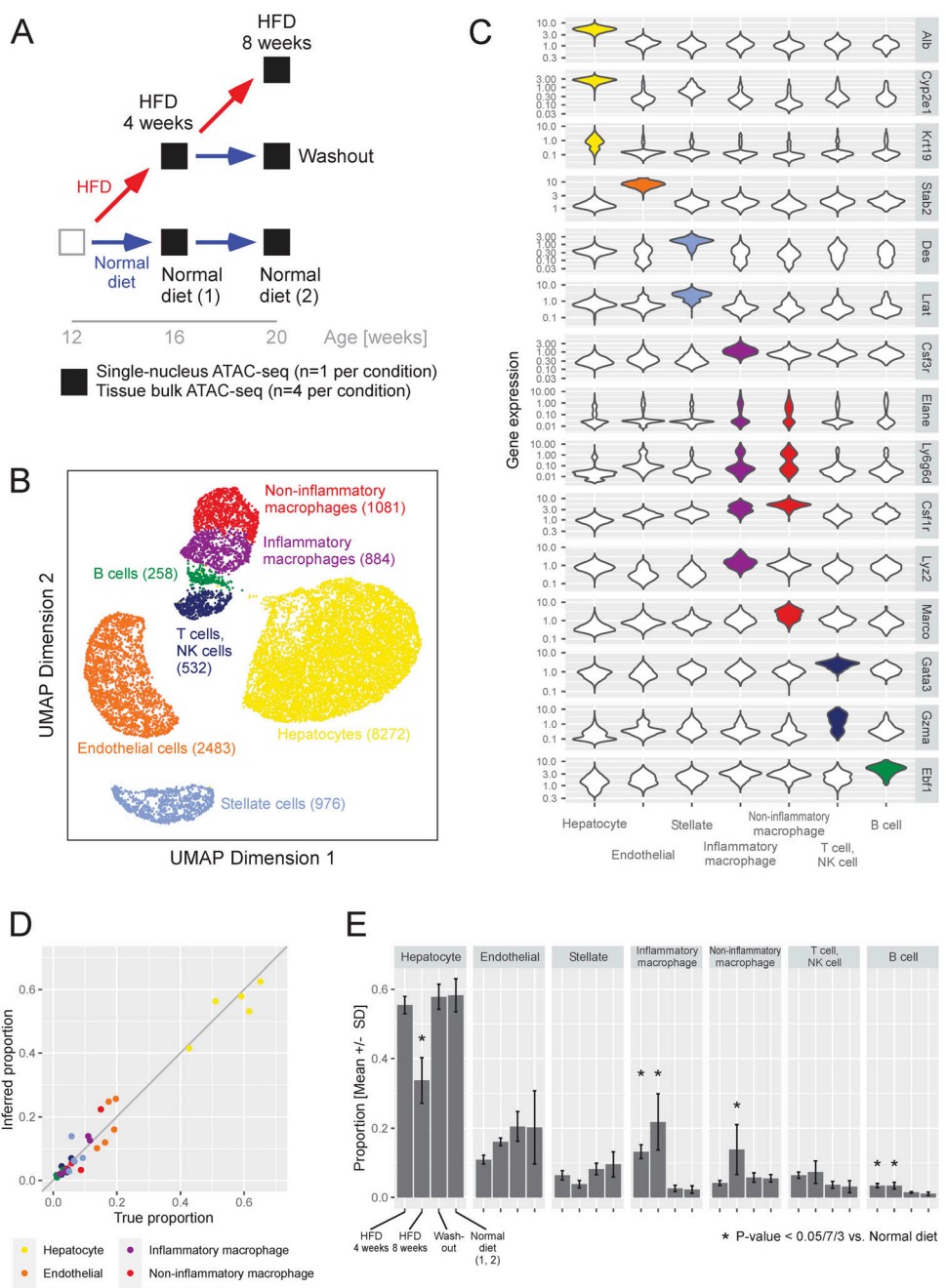

**Figure 1. Cell types observed in the liver and their composition.**
**(A)** Experimental design. **(B)** UMAP plot of the observed nuclei. The number of observed nuclei is shown in parentheses.
**(C)** Expression score of cell-type marker genes in nuclei (vertical axis) is compared between nuclei clusters (horizonal axis) in violin plots. **(D)** Validation of deconvolution based on ATAC-seq using the samples assayed in both single-nucleus and bulk ATAC-seq. The true cell-type proportion counted in snATAC-seq (horizontal axis) is compared with the proportion inferred based on bulk ATAC-seq (vertical axis). **(E)** Proportion of nuclei from a specific cell type (vertical axis) compared across dietary conditions (horizontal axis). The proportion in bulk-tissue ATAC-seq samples was imputed by deconvolution (n = 4 samples per condition). For each cell type, the difference in proportion among dietary conditions was tested by a linear regression where the dependent variable was the imputed composition and the independent variables were indicators of dietary condition. Asterisks (*) indicate statistically significant changes from normal diet under the significance level of 0.05/[7 × 3], adjusted for multiple testing.

such that TFs within a module regulate genes within the same module but do not regulate genes belonging to other modules. In hepatocytes, we identified four modules (Fig 3A). By themselves, modules are simply a grouping of TFs/genes detected by machine learning and carry no inherent biological meaning on their own. We applied gene set enrichment analysis to the genes within a module, without using the information on TFs, to identify biological processes that characterize each module (Fig 3B). In the following, we describe the modules for hepatocytes, endothelial cells (Fig 4), stellate cells (Fig 5), and macrophages (Fig 6). If a biological process was assigned

to a module, we sought the literature that supports the regulation of the biological process by the TFs within the module. The discovered linkages between the TFs and biological processes found support in the literature for five of the seven identified modules, suggesting the validity of our module discovery algorithm.

Module 1 of hepatocytes was characterized by the binding of STAT family TFs (namely, STAT5B, STAT4, and STAT3) and by genes for steroid metabolism.

Module 2 of hepatocytes, Module 1 of endothelial cells, and Module 1 of macrophages were characterized by the binding of AP-1

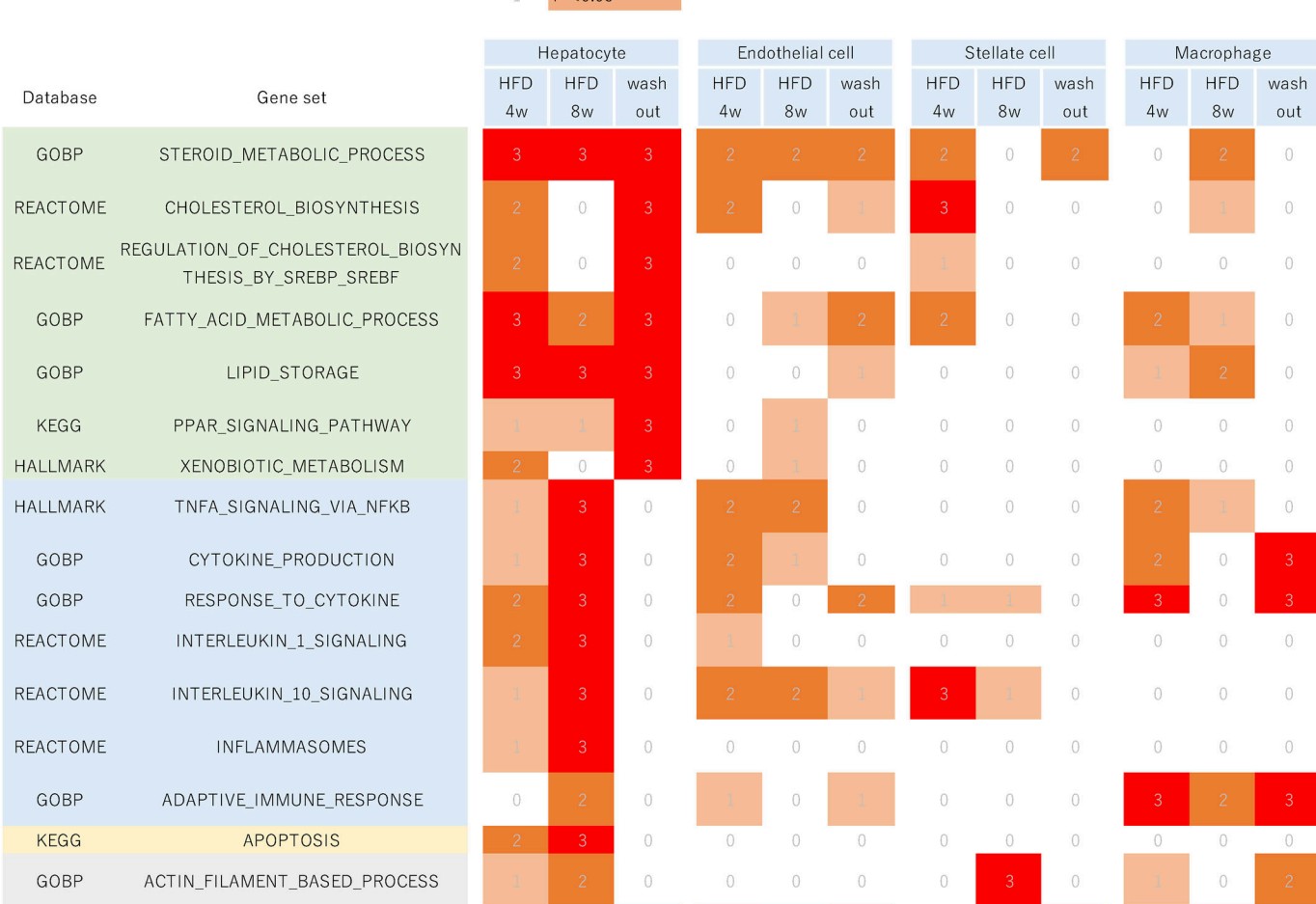

**Figure 2. Biological processes that were differentially expressed under dietary interventions.**
The heatmap illustrates whether a biological process (row) was differentially expressed in a cell type under a dietary intervention compared with the normal diet (column). The color indicates the statistical significance of the gene set enrichment analysis: red, if significant adjusting for multiple testing; brown, if nominally significant at P < 0.01; light brown, if P < 0.05; and white otherwise.

family TFs (namely, FOS, JUND, BACH1, and BACH2), SMARCC1, and NFKB1, along with genes for TNFα signaling via NF-κB. AP-1 TFs respond to cytokine stimuli (Hess et al, 2004). SMARCC1 is a member of the SWI/SNF family of proteins, which functions as a helicase and interacts with NF-κB (Zinzalla, 2016).

Module 3 of hepatocytes was characterized by the binding of the TCF/LEF family TFs (namely, LEF1 and TCF7L1) and by genes in the farnesoid X receptor pathway for bile acid synthesis and genes involved in glutamine family amino acid biosynthesis. The liver lobule, a unit constituting the liver, houses a central vein in its core and a portal triad (portal vein, hepatic artery, and bile duct) in its periphery, with hepatocyte metabolism varying between pericentral and periportal zones. The glutamine family amino acid biosynthesis genes include *Glul* and *Ass1*, which are highly expressed in the pericentral and periportal zones, respectively (Halpern et al, 2017). Moreover, LEF1 TF binds to β-catenin protein, activating the Wnt signaling pathway (Sun & Weis, 2011), and liver-specific β-catenin KO mice exhibited defective cholesterol and bile acid metabolism in the liver (Behari et al, 2010). Wnt-activated genes are also highly expressed in the pericentral zone, suggesting LEF1 TF's role in regulating bile acid metabolism and zonated gene expression in the liver.

Module 2 of endothelial cells was characterized by the DNA binding of AHCTF1 (aka ELYS) and ZNF740 and by genes for

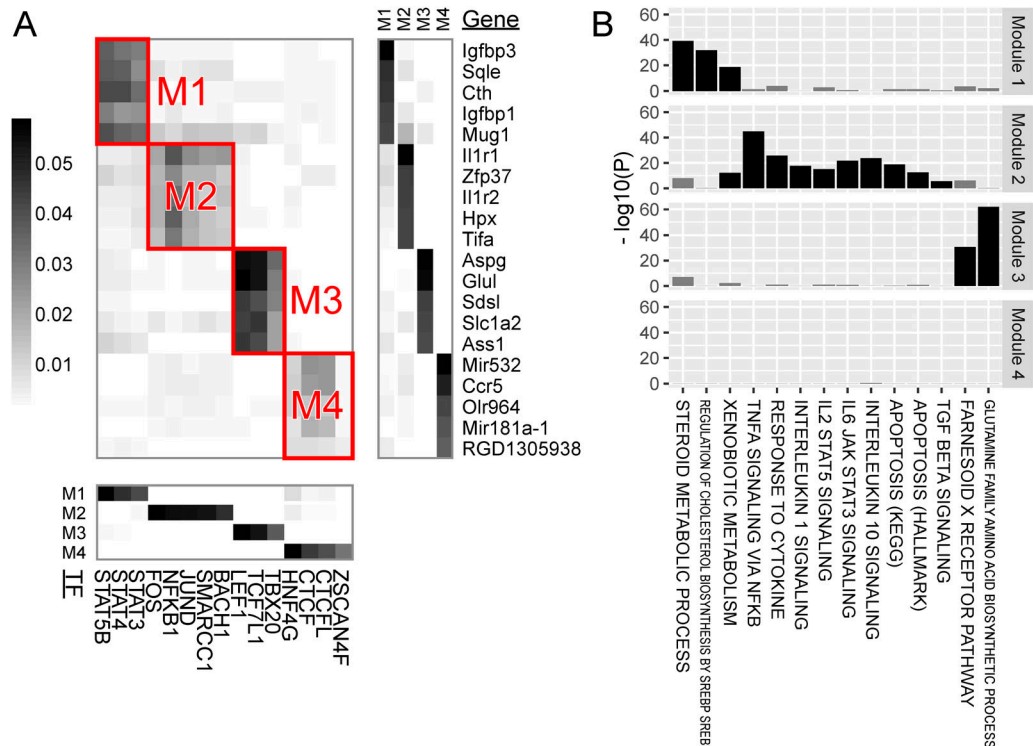

**Figure 3. Major modules of TF regulation in hepatocytes.**
**(A)** The central heatmap visualizes the regulatory impact of TF binding (columns) on genes (rows). The regulatory impact is quantified as the proportion of the variance of a gene that can be explained by a TF and is illustrated according to the coloring in the left bar. Although the impact was computed between all TFs and all genes, only representative TFs and genes are plotted for visibility. Subgroups of TFs/genes demonstrating strong within-group regulation were extracted as modules, which are highlighted by red rectangles. The heatmap at the bottom shows the membership of TFs (columns) to modules (rows). The heatmap on the right displays the membership of genes (rows) to modules (columns). The heatmaps only include representative TFs/genes with membership rankings in the top five and membership weights exceeding half of the top. **(B)** Gene set enrichment analysis for genes belonging to the modules. Excessive overlap between genes belonging to a module and the gene set for a biological process (horizontal axis) was tested, and the *P*-value was plotted (vertical axis). Bars are colored black if statistically significant, adjusting for multiple testing, and gray otherwise. Biological processes that were significant for at least one module are shown.

angiogenesis. AHCTF1 is a DNA-binding protein necessary for mitosis (Rasala et al, 2006). ZNF740 activates angiogenesis in rat pulmonary artery endothelial cells (Yu et al, 2018).

Module 1 of stellate cells was characterized by SOX9 TF binding and genes involved in the semaphorin–plexin signaling pathway.

Module 2 of macrophages was characterized by the binding of Maf family TFs (namely, MAFA, MAFB, and MAFG) and by genes for complement system activation. In *Mafb*-deficient macrophages of mice, C1q production decreased (Tran et al, 2017).

Module 3 of macrophages was characterized by the binding of IRF family TFs (namely, IRF1, IRF2, and IRF7), which regulates immune responses, and by genes associated with angiogenesis. IRF1 contributes to the commitment of pro-inflammatory M1 macrophages, which produce angiogenic stimulators (Chistiakov et al, 2018).

## Core genes in specific biological processes

In the previous section, we discovered major modules of TF regulation and searched databases, such as Gene Ontology, for biological processes characterizing these modules. We defined a biological process by its constituent genes, terming them as a gene set (GS). A GS characterizes a module if it overlaps significantly with the genes within that module. Here, we aim to find "core genes" of a biological process. In preparation, we compute the activity of a GS in individual nuclei as the "GS activity score." The score is obtained by averaging the binding/expression scores of the TFs/genes in the set, or, more precisely, by computing the first principal component. In hepatocytes, regarding the biological process of TNF$\alpha$ signaling via NF-$\kappa$B, the GS activity score was the highest in nuclei from 8 wk of HFD (Fig 7A, top panel).

Core genes are those central to the GS in terms of two connections: co-expression and protein–protein interaction. For co-expression, we first compute the correlation coefficient between the GS activity score and the expression score of a gene (or the binding score of a TF). A core gene is required to have a high absolute value of this correlation coefficient. Furthermore, the total amount of protein–protein interactions between the gene and the GS should be high. Consequently, by plotting the correlation coefficient on the horizontal axis and the total protein–protein interactions on the vertical axis, core genes are expected to appear in either the top left or top right corner. This approach can be used to prioritize known genes in the target GS (Fig 7A, middle panel), and discover novel genes not included in the GS (Fig 7A, bottom panel). In the following, we will explore some of the biological processes identified in the previous section.

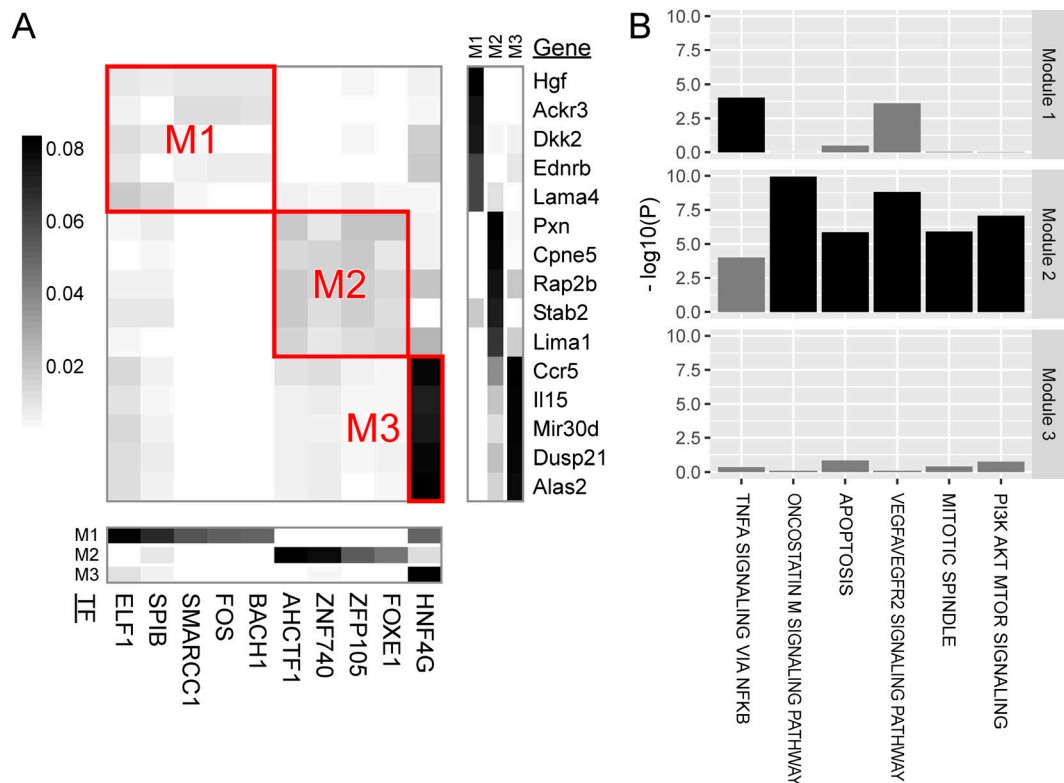

**Figure 4. Major modules of TF regulation in endothelial cells.**
**(A)** Heatmap visualizing the regulatory impact of TF binding (columns) on genes (rows). **(B)** Gene set enrichment analysis for genes belonging to the modules. See legend in Fig 3.

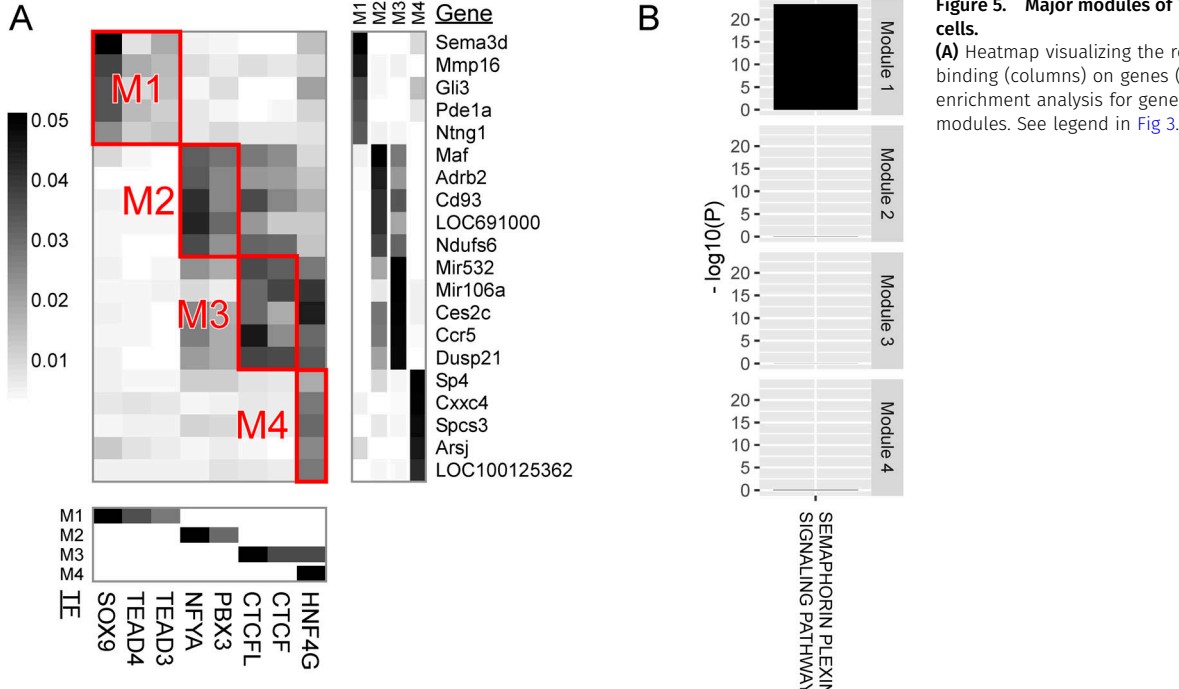

**Figure 5. Major modules of TF regulation in stellate cells.**
**(A)** Heatmap visualizing the regulatory impact of TF binding (columns) on genes (rows). **(B)** Gene set enrichment analysis for genes belonging to the modules. See legend in Fig 3.

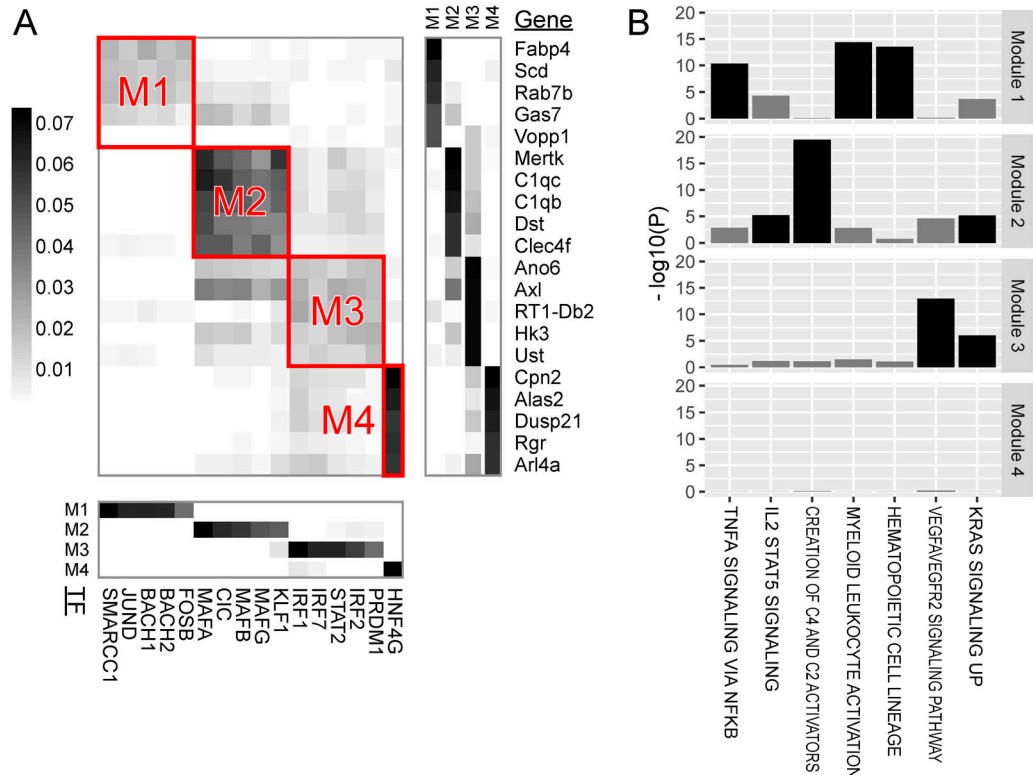

**Figure 6. Major modules of TF regulation in macrophages.**
**(A)** Heatmap visualizing the regulatory impact of TF binding (columns) on genes (rows). **(B)** Gene set enrichment analysis for genes belonging to the modules. See legend in Fig 3.

In hepatocytes, the GS activity score for TNFα signaling via NF-κB (Fig 7A) was the highest after 8 wk of HFD. TFs that had positively correlated binding scores and abundant protein–protein interactions included FOS/JUN and NFKB1/RELA. The positively correlated core genes were *Tnf*, *Nfkb1*, *Bcl2l1*, and *Il1r1*. The negatively correlated core gene was *Cxcl12*.

In endothelial cells, the GS activity score for TNFα signaling via NF-κB (Fig 7B) was higher under HFD conditions. The positively correlated core genes were *Socs3* and *Pecam1*. The negatively correlated core genes were *Tlr2*, *Irs2*, *Tlr4*, *Il15*, *Ccr5*, and *Cxcr4* (receptor for the chemoattractant CXCL12 [Salcedo et al, 1999]).

In macrophages, the GS activity score for TNFα signaling via NF-κB (Fig 7C) was higher after 4 wk of HFD. TFs that had positively correlated binding scores and abundant protein–protein interactions included JUN and KLF6. The positively correlated core genes were *Cd44*, *Fn1*, and *Cxcr4*. KLF6 regulates the switching of macrophages between pro- and anti-inflammatory states (Date et al, 2014). In accordance, *Lyz2* and *Marco*, marker genes for inflammatory and noninflammatory macrophages, respectively, showed a correlation of 0.39 and –0.49, with KLF6 binding in our single-nucleus data.

In hepatocytes, the GS activity score for steroid metabolism (Fig S2A) was the highest after 8 wk of HFD and to a lesser degree after 4 wk of HFD. The positively correlated core genes included *Abcg1*, *Abca1*, and *Scd*. The negatively correlated core genes included *Sqle*, *Hmgcr*, *Apoa1*, *Sult2a1*, *Acox2*, and *Alb*. Among the 15 cytochrome

P450 genes showing a strong negative correlation (correlation less than –0.3 and protein–protein interactions within the top 5%), 13 were oxidoreductases of EC 1.14.14.1. For reference, we labeled the binding of steroid sensing TFs, NR1H2/3 (aka LXRα/β), SREBF1, and SREBF2, which were negatively correlated with the GS expression.

To investigate lobular zonation genes in hepatocytes, the GS was taken from the list of zonation genes in normal mice (q-value <$10^{-6}$ in Table S3 in [Halpern et al, 2017]). The GS activity score showed a modest difference between dietary conditions (Fig S2B), aligning with the concept that zonation is a normal phenomenon unrelated to NAFLD. When classified by the correlation with the GS activity score, the genes with a correlation coefficient >0.3 were all pericentral and those with a correlation coefficient <–0.3 were all periportal, showing that the GS activity score successfully rediscovered the polarity of zonation.

In stellate cells, the GS activity score for semaphorin–plexin signaling was higher at 8 wk of HFD (Fig S2C). The positively correlated core gene was *Nrp2*. The negatively correlated core genes were *Nrp1*, *Plxnd1*, semaphorin genes (*Sema3d*, *Sema3e*, and *Sema6d*), and *Ntn1*.

## Discussion

By performing single-nucleus and bulk ATAC-seq on the liver of rats fed or withdrawn from HFD, we studied the transition in cell-type

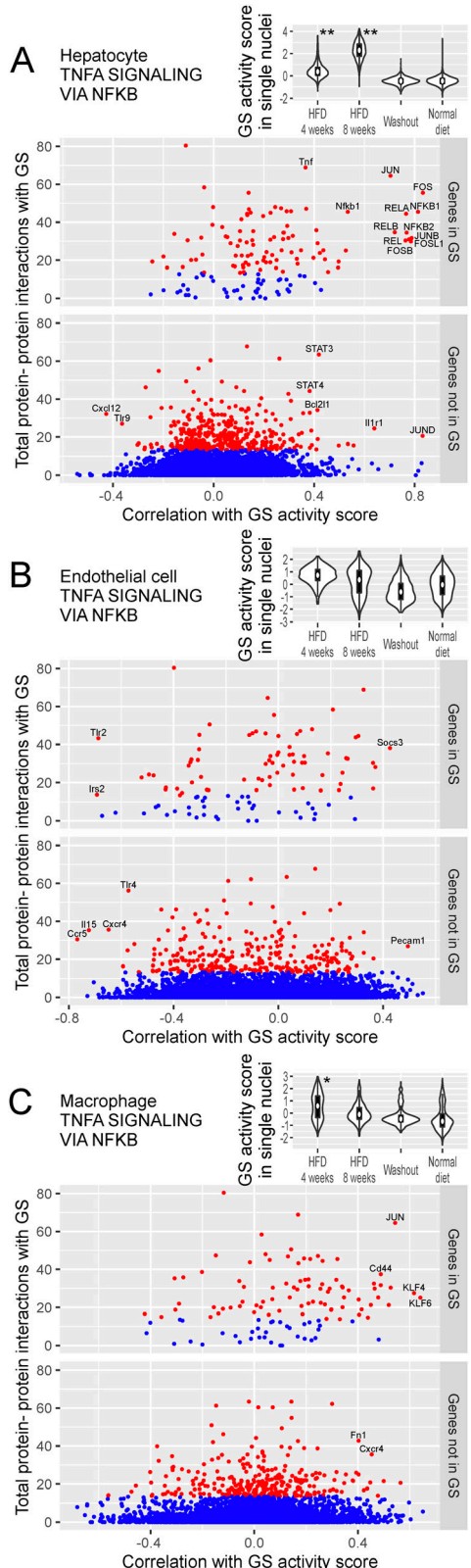

composition of tissues and cell-type-specific gene regulation in a rat model of NAFLD. The accuracy of ATAC-seq–based deconvolution was validated and used to analyze cell-type composition. In accordance with the pathological progression from fatty liver to steatohepatitis that occurred between 4 and 8 wk of the HFD intervention in our rat model, the proportion of inflammatory macrophages dramatically increased. In contrast, after 8 wk of HFD, the proportion of hepatocytes substantially decreased. In the washout condition, where 4 wk of HFD was followed by 4 wk of a normal diet, the cell-type composition reverted to a state similar to that of rats solely fed with the normal diet.

From the global analysis of TF binding and gene expression in single nuclei, we identified major modules of TF regulation (Fig 8), which could be shared among different cell types or be specific to certain cell types. One module shared among hepatocytes, endothelial cells, and macrophages was characterized by the binding of AP-1 TFs and a biological process of TNFα signaling via NF-κB associated with core genes that appear to be cell-type-specific. For many of the modules identified, TFs within a given module were known to regulate biological processes assignable to the genes in the same module, thereby supporting the validity of our module discovery algorithm. Lastly, for the biological processes that unveiled in the TF regulation analysis, by incorporating known protein–protein interactions, we could identify core genes, many of which overlap with genes previously implicated in NAFLD (see below).

We hypothesized that the gene regulatory machinery of a cell is modular and some of the modules would become active when perturbed by HFD. For analogy, a car consists of many machinery parts, and on a cold sunny day, the engine and air conditioning are active but the wiper is not. To discover gene-regulatory modules responsive to an HFD stimulus, we conducted an integrative analysis of TF binding and gene expression in nuclei derived from rats subjected to varying dietary conditions. In each cell type, we applied machine learning to decompose the global variation of TF regulation into major modules, where the TFs within a given module appear to regulate the genes within the same module. In the modules identified as such, the TFs and the function of regulated genes agree with current literature: for example, the AP-1 TFs that regulate inflammation genes (Module 2 of hepatocytes, Module 1 of endothelial cells, and Module 1 of macrophages) are known to respond to cytokine stimuli (Hess et al, 2004), and LEF1 that regulates zonation (Module 3 of hepatocytes) is known to activate Wnt signaling (Sun & Weis, 2011). Previous snATAC-seq studies have

**Figure 7. Core genes for the TNFα signaling via NF-κB processes are identified based on correlation with GS activity score and protein–protein interaction.**
**(A)** Analysis in hepatocytes using the GS for TNFα signaling via NF-κB. In the top panel, the GS activity score in single nuclei (vertical axis) is compared across

dietary conditions (horizontal axis) in violin plots. Box plots within the violins indicate quartiles. The score distribution under a dietary intervention was compared with that under the normal diet condition using the area under the receiver operating characteristic curve: * and ** indicate larger than 0.75 and 0.9, respectively. In the remaining panels, TFs and genes are plotted by their correlation with the GS activity score (horizontal axis) and the total amount of protein–protein interactions with the GS (vertical axis). Points representing TFs/genes having protein–protein interactions within the top 5% are colored red, whereas others are colored blue. The TFs/genes in the GS (middle panel) and not in the GS (bottom panel) are plotted separately; the former exhibit more abundant protein–protein interactions. TFs are labeled in all capital letters. The GS was taken from the MSigDB Hallmark database. Protein–protein interactions were taken from the STRING database (version 11.0). **(B)** Analysis in endothelial cells. **(C)** Analysis in macrophages.

▶▶▶ **Life Science Alliance**

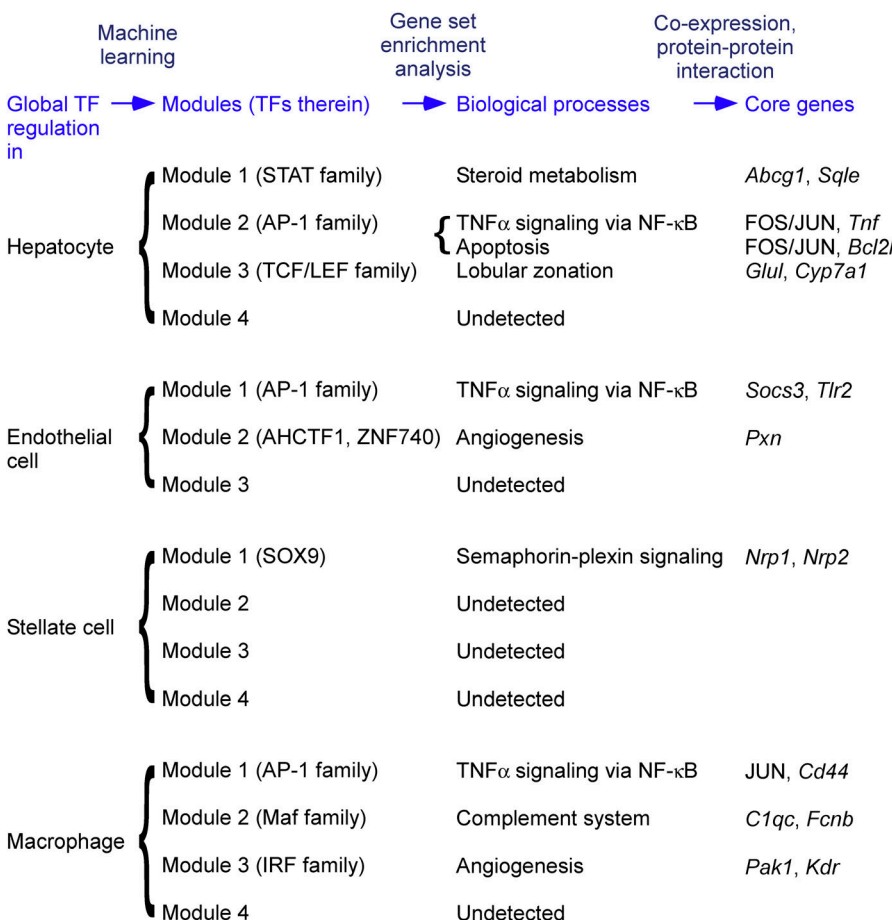

**Figure 8. Modules of global TF regulation, biological processes, and core genes were detected by the data-driven approach.**
See text for the full list of core genes.

characterized the difference between cell types (Cusanovich et al, 2018) using TF binding. In contrast, we illustrated how TFs control the states of cells belonging to one cell type (e.g., the hepatocytes with various degrees of inflammation) possibly in a cell-type-specific manner (e.g., a distinct set of core genes and other target genes associated with AP-1 TFs in different cell types).

A novel method developed in this article is the calculation of the GS activity score in single cells to indicate the activity of a GS, as if it were the expression of a representative gene. In experiments, gene expression can be directly measured, but the activity of a pathway, although of interest, is not directly measurable. One approach could be to take the average expression of genes in that pathway, but this is suboptimal for the following reasons. A simple average would attribute equal weight to both core and peripheral genes in the pathway, thereby amplifying noise from the latter. Moreover, activator and suppressor genes contribute to the pathway in opposing directions. Rather than a simple average, we adopted the first principal component to address these issues. The principal component approach was adapted from the PLAGE program (Tomfohr et al, 2005), which computes pathway activity scores in bulk-tissue specimens. The GS activity score could improve our understanding of biological processes. For example, *Abcg1* and *Sqle* correlated positively and negatively, respectively, with the activity score for the steroid metabolism pathway (Fig S2A). As this score

increased under the HFD, the GS activity represented by the score presumably functions in response to the excess cholesterol contained in the diet. Thus, a positive correlation of *Abcg1* (for lipid efflux) and a negative correlation of *Sqle* (for sterol biosynthesis) are reasonable. Furthermore, the cell-type-specific nature of the GS activity score could reveal cell-type-specific dynamics; the inflammation process peaked under 4 wk of HFD in macrophages and under 8 wk of HFD in hepatocytes (Fig 7). The GS activity score can also be applied to single-cell and single-nucleus RNA-seq data.

Another novel method introduced in this article is the identification of core genes of biological processes. Given a GS comprising known genes for the process, we identified core genes as those exhibiting a strong correlation with the GS activity score and possessing abundant protein–protein interactions with the GS. Evidence for the former was derived from single-nucleus experiment data, and for the latter from a public database. We searched the literature for indications that core genes cause NAFLD or serve as biomarkers for NAFLD, and found such evidence for a significant portion of the core genes (see below): for the TNFα signaling via NF-κB process, four of five in hepatocytes, four of eight in endothelial cells, and one of three in macrophages; for steroid metabolism, three of nine in hepatocytes; and for semaphorin–plexin signaling, three of seven in stellate cells. The substantial overlap between our current findings and known NAFLD genes serves as a

proof-of-concept for our data-driven approach, which uses snATAC-seq data from a rat model of dietary intervention as input, then extracts modules, biological processes, and finally, core genes. Regarding TNFα signaling via NF-κB in hepatocytes, in animal studies, *Tnf* KO mouse was protected from NAFLD (Kakino et al, 2017), *Nfkb1* KO mouse exacerbated NAFLD (Locatelli et al, 2012), and *Il1r1* KO mouse had less hepatic triacylglycerol (Roos et al, 2009). CXCL12 (aka SDF1) induced migration of stellate cells (Sawitza et al, 2009), endothelial cells (Salcedo et al, 1999), and CD4⁺ T cells (Boujedidi et al, 2014). In human biomarker studies, higher circulating TNFα level was associated with NAFLD (Potoupni et al, 2021) and future NAFLD (Seo et al, 2013). In NAFLD patients, soluble IL1R1 was higher (Ajmera et al, 2017). Regarding TNFα signaling via NF-κB in endothelial cells, in animal studies, *Pecam1* KO mouse developed nonalcoholic steatohepatitis (Goel et al, 2007), whereas *Tlr4* KO mouse (Zhang et al, 2020) and *Il15* KO mouse (Cepero-Donates et al, 2016) were protected from NAFLD. *Ccr5* antagonist alleviated NAFLD in mice (Pérez-Martínez et al, 2014). Regarding TNFα signaling via NF-κB in macrophages, *Cd44* KO mouse had less liver injury under a NAFLD-inducing diet, and NAFLD patients had elevated serum-soluble CD44 (Patouraux et al, 2017). Regarding steroid metabolism in hepatocytes, *Scd1* KO mouse exacerbated NAFLD (Li et al, 2009), and *Acox2* KO mouse developed NAFLD (Zhang et al, 2021). NAFLD patients exhibited lower serum APOA1 (Yang et al, 2016; Chen et al, 2022). Regarding semaphorin–plexin signaling in stellate cells, in animal studies, *Nrp2* conditional KO reduced liver fibrosis in mice (Rigotti et al, 2022 *Preprint*). *Nrp1* KO attenuated cell motility and overexpression increased motility (Cao et al, 2010). *Sema3e* KO alleviated liver fibrosis in mice (Yagai et al, 2014).

In the pursuit of potential drug targets or biomarkers for NAFLD, we sought proteins whose subcellular localization is extracellular or plasma membrane proteins (Binder et al, 2014) among the genes strongly up-regulated under dietary intervention (Table S2) or the core genes positively correlated with NAFLD-related activity. Previous reports on NAFLD biomarkers in humans are cited. For hepatocytes, we identified *Il1r1* (Ajmera et al, 2017), *Il1r2*, and *Tnf* (Potoupni et al, 2021). For endothelial cells, *Hgf* was found (Koutsogiannis et al, 2010; Krawczyk et al, 2017; An et al, 2019 *Preprint*). *Pecam1* was listed for both endothelial and stellate cells. For stellate cells, we found *Faslg* (Page et al, 2013) and *Nrp2*. For macrophages, we identified RT1-Db1, *Cd44* (Patouraux et al, 2017), *Cxcr4*, and *Fn1*. We also searched DGIdb (2022–Feb) for existing drugs that could potentially be repurposed, finding antagonist/antibody/inhibitor of up-regulated or positively correlated genes, *Bcl2l1*, *Cxcr4*, *Hgf*, *Il1r1*, and *Tnf*, and agonist of down-regulated or negatively correlated genes, *Ccr5*, *Cxcr4*, *Tlr2*, and *Tlr4* (Table S3). *Cxcr4* was positively correlated with TNFα signaling via NF-κB in macrophages but negatively correlated in endothelial cells.

This study has several limitations. Firstly, it was difficult to distinguish TFs with very similar motifs. A specific TF was presumed to bind a specific site if the chromatin was open near the site and if the DNA sequence of the site matched the known TF-binding motif. However, instead, a different TF with a similar motif could have bound at that location. Secondly, the interpretation of gene expression requires caution. Gene expression can be quantified by the amount of either (A) mRNA transcription per unit of time, (B) mRNA in the nucleus, (C) mRNA in the cell, (D) protein translation per unit of time or (E) protein. Bulk tissue measurements are conducted by ATAC-seq (A), RNA-seq (C), Ribo-seq (D) or Western blotting (E), whereas single-nucleus/cell measurements are performed by snATAC-seq (A), single-nucleus RNA-seq (B) or single-cell RNA-seq (C). Whereas the gene expression score in this study, which is akin to A, might reflect the state of gene regulation, it could differ from E, which reflects the metabolic state, and the fold-change value would differ among these types of gene expression. Lastly, the TF regulation might differ between rat, mouse, and human.

To discover TF modules, we used machine learning, the performance of which can be influenced by the selection of hyperparameters. If a hyperparameter allows the machine learning model to be excessively complex and flexible relative to the available data, the model may erroneously interpret noise as a signal, leading to overfitting. We chose hyperparameters based on previous empirical validations (for the number of candidate regulators at each node in GENIE3) or residual error (for the number of factors in NMF), but we did not extensively explore different hyperparameters. We evaluated the chosen hyperparameters post hoc. Across a range of hyperparameters, we randomly divided the hepatocytes into two halves, performed TF module discovery in each set, and assessed consistency. We did not observe overfitting in the hyperparameter for the number of candidate regulators (Fig S3A) nor in the hyperparameter for the number of factors in NMF (Fig S3B). Nevertheless, an assessment using data from an independent experiment would be beneficial.

By performing snATAC-seq in a rat model of NAFLD and employing novel statistical methods, we elucidated a global picture of in vivo TF regulation in each cell type as a set of modules and discovered core genes for NAFLD-relevant biological processes. The core genes can serve as novel candidates for NAFLD causation and our statistical methods can be applied to elucidate TF regulation in biological systems beyond NAFLD.

# Materials and Methods

### Animal experiment

We used male rats of the SHR/Izm strain, which is an inbred genetic model of essential hypertension. The strain was supplied by Japan SLC. Rats were weaned at 4 wk after birth and placed on a normal diet (SP diet; Funabashi Farm). According to the study design, adult rats were either maintained on the normal diet or switched to a high-fat atherogenic diet (24% fat, 15% protein, 5% cholesterol, 2% cholic acid). Compared with models based on nutritional deficiencies (such as the methionine and choline-deficient diet) or chemically induced models (such as carbon tetrachloride), HFD models more closely resemble human NAFLD, both pathophysiologically and phenotypically (Herck et al, 2017). The SHR strain can mimic NAFLD under a hypertension background. Rats were euthanized under pentobarbital anesthesia (200 mg/kg via intraperitoneal infusion), and the organs were excised and immediately frozen at –70°C. We used the liver's largest lobe (left lateral lobe) for pathology imaging and the right segment of the second-largest lobe (left median lobe) for genomics analysis. The measured outcomes were genomic and biochemical assays of the liver,

pathology imaging of the liver, biochemical assays of the serum, organ and body weight, and blood pressure. The experimental unit was the individual animal. No animal was excluded during the experiment. The allocation of animals to dietary conditions was not randomized. Although we controlled the dietary conditions, other potential confounders were not accounted for. Group allocation was known to all experimenters except for the pathologist, who examined unlabeled preparations. All animal experiments conformed to the Guidelines for Animal Experiments of the National Center for Global Health and Medicine (NCGM) and were carried out in the animal facility of NCGM. The animal experiments and the protocol were registered and approved by the Animal Research Committee of NCGM (permission number 21009).

### Library preparation and sequencing for snATAC-seq

Nuclei were isolated from flash-frozen liver tissue according to the 10x Genomics protocol CG000212 for samples m154207 and m154211. For samples m167108, m167203, and m168101, we used the Minute Single Nucleus Isolation Kit for Tissues/Cells (SN-047; Invent Biotechnologies). For each sample, the sequencing library was prepared with the Chromium Next GEM Single Cell ATAC Reagent Kits v1.1 (10x Genomics, CG000209) and sequenced on the NovaSeq 6000 sequencer (Illumina).

### Library preparation and sequencing for bulk-tissue ATAC-seq

Bulk-tissue ATAC-seq was performed according to the Omni-ATAC protocol (Corces et al, 2017) using flash-frozen liver tissue. The library was sequenced on the NextSeq sequencer (Illumina).

### Measuring triglycerides in the liver tissue

Triglycerides in the liver tissue were measured enzymatically using the Adipogenesis Colorimetric/Fluorometric Assay Kit (K610-100; BioVision). Tissue homogenate was obtained by dispersing 10 mg tissue in 1 mL lipid extraction solution under 2,200 rpm for 1 min, using a handheld disperser (PT 1300 D; Kinematica Inc.). For fatty liver tissues, we prepared a 1/3 dilution of the homogenate by adding more lipid extraction solution and examined both the original homogenate and the diluted sample. Triglycerides were measured in a colorimetric assay according to the manual. The triglyceride quantity in mole units was converted to gram units by multiplying by 885.7.

### Measuring total cholesterol in the liver tissue

Total cholesterol in the liver tissue was measured enzymatically using the Total Cholesterol and Cholesteryl Ester Colorimetric Assay Kit II (K623-100; BioVision). Tissue homogenate was obtained by dispersing 15 mg tissue in 600 $\mu$l $CHCl_3$:IPA:NP-40 (7:11:0.1) at 2,200 rpm for 1 min, using a handheld disperser (PT 1300 D; Kinematica Inc.). The homogenate was then centrifuged for 5 min at 15,000$g$. Afterward, we transferred 200 $\mu$l of the liquid phase to a new tube, dried it with airflow-in at 40°C for 10 min using a centrifugal concentrator (CC-105; TOMY Digital Biology), and dissolved the dried lipids with 400 $\mu$l Cholesterol Assay Buffer. Total cholesterol was measured in a colorimetric assay according to the manual.

### Data cleaning for snATAC-seq

Data cleaning was conducted separately for each sample. ATAC-seq fragments were assigned to individual nuclei, and only nuclei of good quality were retained, as described below. In principle, one nucleus and one barcoded bead are contained in one droplet of the chromium assay, thus ATAC-seq fragments originating from the same nucleus can be identified by the shared unique barcode. Using Cell Ranger ATAC software (version 2.0.0), we mapped the paired-end reads to the mRatBN7.2/rn7 rat reference genome and identified ATAC-seq fragments, which were assigned to unique barcodes.

Occasionally in the chromium assay, more than one barcoded bead can be contained in a single droplet. In such cases, an excess of ATAC-seq fragments would be commonly assigned to the barcodes. We detected these instances and unified and renamed the barcodes using the bap2 software (version 0.6.7) (Lareau et al, 2020). In subsequent analyses, we could assume that one barcode corresponds to one droplet.

We managed the snATAC-seq data using the ArchR package (version 1.0.1) (Granja et al, 2021) of the R software (version 4.1.1). ATAC-seq fragments from the autosomes and chromosome X were retained. We kept nuclei that had abundant and enriched fragments around transcription start sites (TSS) as follows: ≥1,000 fragments within 1 kb of TSS, the proportion of fragments within 1 kb of TSS being ≥7.5% (for samples m154207, m154211, and m167203) or ≥15% (for samples m167108 and m168101), and Tn5 insertions (which occur at both ends of a fragment) within 25 bp of TSS occurring ≥4 times more frequently than insertions at 1,900–2,000 bp from TSS.

For each nucleus, the proportion of fragment counts from a chromosome (e.g., chromosome 1) is approximately constant, being proportional to the chromosome length. However, this could deviate if a droplet is contaminated by free DNA from broken nuclei (Orchard et al, 2020). Therefore, we represented the proportion by a matrix indexed by nuclei (rows) and chromosomes (columns). To standardize the matrix, we subtracted column-wise averages and divided by the SD. By taking the sum of the absolute values of a row, we defined the deviation score of a nucleus. To exclude contaminated droplets, we discarded nuclei with a score of ≥40.

Because Tn5 inserts less efficiently into nucleosome-associated DNA, ATAC-seq fragment size distribution exhibits periodicity (Cusanovich et al, 2018). Using the program provided by Cusanovich et al (2018) (https://atlas.gs.washington.edu/mouse-atac/docs/), we computed, for each nucleus, the spectral densities for the 100–300-bp range, which corresponds to the pitch of nucleosomes. To exclude nuclei with aberrant chromatin states, we discarded those with scores below the first or above the 99th percentile.

When two or more nuclei are contained in one droplet, known as a multiplet, those nuclei share the same barcode, violating the one-nucleus-to-one-barcode assumption. We removed the barcodes for multiplets in two ways. First, we inferred the number of homologous chromosomes from which the ATAC-seq fragments assigned to a barcode were derived. This number is analogous to ploidy, but differs because the container is a droplet, not a cell. By counting overlaps among the fragments, the number of homologous chromosomes was estimated to be either 2$n$, 4$n$ or 8$n$ using the R package scPloidy (version 0.3.0). Because most of the hepatocytes in rat liver are either diploids or tetraploids (Katsuda et al, 2020) and other cell types in the

 **Life Science Alliance**

liver are diploids, droplets with 8$n$ were regarded as potential multiplets and discarded. Second, we detected doublets formed by two heterogeneous cell types using ArchR. Droplets with a doublet enrichment score >10 were discarded.

## Chromatin accessibility peak discovery and nucleus clustering for snATAC-seq

The Tn5 insertions observed in ATAC-seq occur at open chromatin regions, such as promoters and enhancers, and the genome-wide profile of Tn5 insertions characterizes the distinct cell types. We performed nucleus clustering based on the similarity of Tn5 insertions. Firstly, we represented the Tn5 insertions in snATAC-seq as a "tile matrix": a binary matrix indexed by tiles of 500-bp width across the genome (rows) and nuclei (columns), coding for the presence (1) or absence (0) of Tn5 insertions. Using ArchR, we generated a matrix with 5,230,329 rows and 14,615 columns. Subsequently, we grouped nuclei based on the similarity of the corresponding columns (see the following section). For each group, pseudo-bulk ATAC-seq datasets were generated, and Tn5 insertion peaks were computed using ArchR and the MACS2 programs (version 2.1.4) (Zhang et al, 2008). We identified 599,035 chromatin accessibility peaks with a width of 501 bp. The fraction of reads in peaks was 0.444 (sample m154207), 0.407 (m154211), 0.541 (m167108), 0.446 (m167203), and 0.557 (m168101). The proportion of reads that aligned within 50 bp of TSS was 0.0286 (sample m154207), 0.0224 (m154211), 0.0604 (m167108), 0.0340 (m167203), and 0.0646 (m168101).

Secondly, we represented the Tn5 insertions as a "peak matrix": a binary matrix indexed by peaks (rows) and nuclei (columns), coding for the presence or absence of Tn5 insertions. Again, we grouped nuclei based on the similarity of the corresponding columns. The nucleus clustering based on the peak matrix was used in downstream analysis, as it is likely more accurate than the clustering based on the tile matrix.

## Algorithm for dimensionality reduction and nucleus clustering

The tile and peak matrices are large binary matrices indexed by genomic positions (rows) and nuclei (columns). Because the column vectors are extremely high-dimensional, we employed dimensionality reduction to effectively analyze the similarity among nuclei. We used latent semantic indexing (Berry & Browne, 2005) and singular value decomposition (SVD) for this purpose. Dimensionality reduction was performed on all samples combined. As experimental conditions, including the Tn5:DNA ratio, may vary among samples, we introduced a step for sample batch correction. We applied the same algorithm to both the tile and peak matrices.

The input binary matrix ($f_{ij}$) with $n$ columns was treated as the term–document matrix in the latent semantic indexing literature. We defined the inverse document frequency (IDF) as

$$g_i = \log_2\left(\frac{n}{\sum_j f_{ij}}\right).$$

For the $i$th row of the matrix weighted by IDF, the variance becomes

$$g_i^2 \left(\frac{\sum_j f_{ij}}{n}\right)\left(1 - \frac{\sum_j f_{ij}}{n}\right),$$

which ranges between zero and one. To limit noise, we retained rows with an IDF-weighted variance >0.4 for the tile matrix and >0.1 for the peak matrix. We defined the document normalization factor as

$$d_j = \frac{1}{\sqrt{\sum_i f_{ij}^2}}.$$

We applied sample batch correction independently to the rows of the matrix. For row $i$, the normalized elements $d_j \cdot f_{ij}$ vary among nuclei $j$, but the average across many nuclei in a sample is anticipated to be approximately constant. The difference in the averages among samples was regarded as a batch effect and was corrected by scaling. For nucleus $j$ belonging to sample $s$, the correction factor is

$$\gamma_{ij} = \frac{\text{Mean}\left[\left\{d_{j'} \cdot f_{ij'} : 1 \le j' \le n\right\}\right]}{\text{Mean}\left[\left\{d_{j'} \cdot f_{ij'} : j' \text{ belongs to sample } s\right\}\right]}. \qquad (1)$$

If the denominator equals zero, $\gamma_{ij}$ is set as one. The above correction requires no knowledge of cell types. However, if cell type classification of nuclei is available, the correction can be refined. For nucleus $j$ of cell-type $c$ belonging to sample $s$, the correction factor becomes

$$\gamma_{ij} = \frac{\text{Mean}\left[\left\{d_{j'} \cdot f_{ij'} : j' \text{ is cell type } c\right\}\right]}{\text{Mean}\left[\left\{d_{j'} \cdot f_{ij'} : j' \text{ is cell type } c \text{ and belongs to sample } s\right\}\right]}. \qquad (2)$$

As we had no prior knowledge of cell type, we first applied the batch correction in Equation (1) and then inferred a coarse cell-type classification. Using the coarse classification, we applied the batch correction in Equation (2). To avoid the inflation of $\gamma_{ij}$ under a small denominator, for each nucleus $j$, we computed the square of the third quartile of $\gamma_{ij}$ divided by the second quartile, and imposed this as the upper limit.

Dimensionality reduction and nucleus clustering were performed in the following steps:

(1) Compute the sample batch correction factor $\gamma_{ij}$ using Equation (1).
(2) Compute the matrix ($g_i \cdot d_j \cdot f_{ij} \cdot \gamma_{ij}$) and centralize each row by subtracting the average.
(3) Compute the truncated SVD, $UDV^T$, for the 30 largest singular values using the R package irlba (version 2.3.3) (Baglama & Reichel, 2005).
(4) Take the truncated SVD for the two largest singular values, project the nuclei to the $VD$ coordinates, and detect and discard outlier nuclei. Outliers were detected based on the sum of the squared distance to the five nearest neighbors.
(5) Take the truncated SVD for the non-noise singular values, and project the nuclei to the $VD$ coordinates. In our data, as the 11th and later singular values bottomed out and the

corresponding columns of $V$ appeared randomly distributed, we took the 10 largest singular values.

(6) Identify clusters of nuclei by computing the shared nearest-neighbor graph with the R package Seurat (version 4.0.0) (Hao et al, 2021) and optimizing the modularity with the Leiden algorithm (version 0.8.4) (Traag et al, 2019). By merging the clusters, define coarse clustering that clearly separates in the $VD$ coordinates. In our data, the coarse clusters corresponded to hepatocytes, stellate cells, endothelial cells, and white blood cells.

(7) Compute the sample batch correction factor $\gamma_{ij}$ using Equation (2), incorporating the coarse clustering of nuclei.

(8) Compute the matrix $(g_i \cdot d_i \cdot f_{ij} \cdot \gamma_{ij})$ and centralize each row by subtracting the average.

(9) Compute the truncated SVD, $UDV^T$, for the 30 largest singular values.

(10) Take the truncated SVD for the non-noise singular values, and project the nuclei to the $VD$ coordinates. We took the eight largest singular values.

(11) Identify clusters of nuclei through shared nearest-neighbor modularity optimization.

## Cell type assignment for snATAC-seq

For each nucleus cluster obtained above, we assigned a cell type based on chromatin accessibility at known marker genes. For each cluster, chromatin accessibility at the TSS and gene body of the genes was visually inspected using the browser of ArchR. We used the following marker genes in the rat:

· *Alb*, *Apoc3*, *Cyp2e1*, and *Cyp2f4* for hepatocyte
· *Epcam* and *Krt19* for cholangiocyte
· *Stab2* for endothelial cell
· *Dcn*, *Des*, *Lrat*, and *Acta2* for stellate cell
· *Csf3r* for granulocyte
· *Elane*, *Ly6g6c*, *Ly6g6d*, *Ly6g6e*, and *Mpo* for neutrophil
· *Csf1r* for macrophage (including Kupffer cell)
· *Lyz2* for inflammatory macrophage
· *Marco* for noninflammatory macrophage
· *Cd3g* and *Gata3* for T cell
· *Gzma* and *Prf1* for cytotoxic T cell and NK cell
· *Ebf1* for B cell

## Data cleaning for bulk ATAC-seq

In accordance with the ATAC-seq data-processing workflow (Reske et al, 2020), we trimmed reads, mapped reads to the reference genome, eliminated mitochondrial DNA reads, subsampled to equalize the library complexity, and removed PCR duplicates.

## Inferring cell-type composition of bulk tissue samples

We inferred the cell-type composition of liver tissue samples by comparing their bulk ATAC-seq data with the reference data generated by snATAC-seq. As references, pseudo-bulk ATAC-seq data of pure cell types were computed from snATAC-seq results. We used a combination of several published deconvolution methods. Let the indices be $i$ for a chromatin accessibility peak, $j$ for a cell

type, and $k$ for a sample, either assayed in bulk, in single nuclei or both.

By summing the Tn5 insertion counts of snATAC-seq over $a_{jk}$ nuclei of cell type $j$ from sample $k$, we obtain Tn5 insertion count $u_{ijk}$ at peak $i$. The counts across all peaks are represented as a vector $\boldsymbol{u}_{jk}$ in bold font, in which the index $i$ is dropped. We quantile normalize the vector $\boldsymbol{u}_{jk}$ and obtain vector $\boldsymbol{v}_{jk}$. Although quantile normalization is not a linear transformation, we can regard the total number of nuclei in $\boldsymbol{v}_{jk}$ as

$$b_{jk} = \frac{\sum_i v_{ijk}}{\sum_i u_{ijk}} \cdot a_{jk}.$$

We factorize the number as

$$b_{jk} \approx c_j \cdot d_k,$$

where $c_j$ represents the number of nuclei included in the normalized ATAC-seq library for pure cell type $j$ and $d_k$ adjusts for the experimental batch effect between samples. Specifically, we solve a linear regression whose dependent variable is the logarithm of $b_{jk}$ and independent variables are the indicator functions of $j$ and $k$, and take the exponential of the obtained coefficients. For the solution to be unique, we set $d_1 = 1$. We denote $C$ as the diagonal matrix with $c_j$ on the diagonal. The values of $c_j$ for hepatocytes were halved compared with the values for other cell types because of the higher ploidy of hepatocytes.

We used the DeconPeaker software (version 1.0) (Li et al, 2020) to select the peaks (rows) that distinguish cell types. DeconPeaker chooses these peaks based on $t$-tests while minimizing the conditional number. Let $\tilde{\boldsymbol{v}}_{jk}$ be the vector truncated from $\boldsymbol{v}_{jk}$ accordingly. The vectors $\tilde{\boldsymbol{v}}_{jk}$ across the samples from the same cell type $j$ are approximately equal, and their average forms the reference vector $\boldsymbol{w}_j$ for chromatin accessibility:

$$w_{ij} = \mathrm{mean}_k\left\{\tilde{v}_{ijk}\right\}.$$

$W$ represents the matrix whose columns are $\boldsymbol{w}_j$. By regarding the samples as replicates, the error variance for peak $i$ was estimated as

$$e_i^2 = \mathrm{mean}_j\left\{\mathrm{var}_k\left\{\tilde{v}_{ijk}\right\}\right\}.$$

For samples assayed in snATAC-seq, we can generate pseudo-bulk chromatin accessibility for mixtures based on $W$. For sample $k$, the proportion of cell type $j$ is $a_{jk}/(\sum_{j'} a_{j'k})$, and the mixture becomes

$$\boldsymbol{x}_k = \sum_j \frac{a_{jk}}{\left(\sum_{j'} a_{j'k}\right)} \cdot \frac{1}{c_j} \cdot \boldsymbol{w}_j.$$

Here, the vector is scaled to one nucleus (under the experimental condition of sample $k$ = 1). In other words, each element is the average count of Tn5 insertions at the site in one nucleus.

The bulk ATAC-seq results of mixture samples are quantile normalized and represented as vector $\boldsymbol{y}_k$ for sample $k$. The elements $y_{ik}$ are taken only for the peaks $i$ that were selected by DeconPeaker. Five samples were assayed both in bulk and in single nuclei, yielding vectors $\boldsymbol{x}_k$ and $\boldsymbol{y}_k$. Although, in principle, the two vectors should be

proportional, the ratio $y_{ik}/x_{ik}$ actually varies by peaks, because of measurement differences between different assay technologies. Correction of between-peak variability can improve deconvolution (Jew et al, 2020). We compute the correction factor as

$$f_i = \exp\left(\text{mean}_k\left\{\log\left(\frac{y_{ik}}{x_{ik}}\right)\right\}\right).$$

To avoid overcorrection, we cap $f_i$ to range between $Q_1{}^2/Q_2$ and $Q_3{}^2/Q_2$, where $Q_1, Q_2, Q_3$ are the quartiles of the original value. We represent by $F$ the diagonal matrix with $f_i$ in the diagonal. After correction, a bulk ATAC-seq experiment $\mathbf{y}_k$ becomes $F^{-1}\mathbf{y}_k$. Note that the vector is scaled to one nucleus.

For deconvolution of a bulk sample $k$, we first compute $\boldsymbol{\theta}_k$ that approximates

$$F^{-1}\mathbf{y}_k \approx W\boldsymbol{\theta}_k,$$

under $\theta_{jk} \geq 0$. Using robust linear regression (Teschendorff et al, 2017), we solve

$$F^{-1}\mathbf{y}_k = W\boldsymbol{\theta}_k + \boldsymbol{\varepsilon},$$

$$\varepsilon_i \sim N\left(0, e_i^2\sigma^2\right),$$

where $\sigma^2$ is the parameter for error variance. Peaks with large variability $e_i^2$ are allowed to have a larger variance (Wilson et al, 2019). To avoid overcorrection, we cap $e_i^2$ to range between $Q_1{}^2/Q_2$ and $Q_3{}^2/Q_2$, where $Q_1, Q_2, Q_3$ are the quartiles of the original value. The negative elements of the estimate $\hat{\boldsymbol{\theta}}_k$ are set to zero. When computing $\boldsymbol{\theta}_k$, the dependent variable $F^{-1}\mathbf{y}_k$ was scaled to one nucleus, and the regressors $\mathbf{w}_j$ were scaled to $c_j$ nuclei. We multiply $\hat{\boldsymbol{\theta}}_k$ by $C$ and obtain $C\hat{\boldsymbol{\theta}}_k$, which is the fractional number of nuclei from each cell type included per nucleus of sample $k$, which, in spirit, is the cell-type composition. Although the sum of the elements in $C\hat{\boldsymbol{\theta}}_k$ was approximately equal to one, to be precise, we rescaled to obtain the cell-type composition for sample $k$ as

$$\frac{C\hat{\boldsymbol{\theta}}_k}{\sum_j c_j\hat{\theta}_{jk}}.$$

### Comparing cell-type composition among dietary conditions

For each cell type, the imputed composition of bulk samples was compared among dietary conditions. We performed multiple regression where the dependent variable was the imputed composition and the independent variables were indicators of dietary condition, with the normal diet serving as the reference. We applied Bonferroni's correction for multiple testing of seven cell types and three conditions (apart from the normal diet), setting the significance level to $0.05/[7 \times 3]$.

### Quantifying gene expression in snATAC-seq

For each nucleus, the expression level of a gene was inferred from the number of Tn5 insertions in the gene body and vicinity. The targeted biological phenomenon is the chromatin opening of the gene rather than the quantity of mRNA in a nucleus or cell. Using ArchR, we computed the Gene Score Matrix, which is indexed by genes (rows) and nuclei (columns) and represents the insertion counts. As the matrix is analogous to the count matrix obtained in single-cell RNA-seq, we applied the library size normalization of the R package Linnorm (version 2.16.0) (Yip et al, 2017) and the imputation of the R package SAVER (version 1.1.2) (Huang et al, 2018). Before imputation, 47% of the matrix elements were zero, and post-imputation, 0.2% were set to the lowest value (0.001). We excluded low-expression genes, where the lowest value occurred in ≥1% of nuclei. We took the $\log_2$ of the matrix and applied quantile normalization. The obtained matrix was used for inspecting cell-type marker genes. We applied SVD and subtracted the first two components from the matrix, which captured the sample batch effect. Although the two components also captured major cell-type differences, we anticipated that the removal would not harm downstream analysis, which compares nuclei of the same cell type between samples.

### Quantifying TF binding in snATAC-seq

If the DNA sequence of a chromatin accessibility peak includes the motif of a TF, Tn5 insertion in the peak suggests the binding of the corresponding TF. For each nucleus, by aggregating over all peaks, we can infer the genome-wide binding of the TF. For each TF, the enrichment of its motif in the peaks was quantified by the chromVAR (Schep et al, 2017) deviation z-score in ArchR. The CIS-BP database of TF-binding motifs (Weirauch et al, 2014) was used. In the same way as we did for the Gene Score Matrix, we applied quantile normalization to the Motif Matrix and subtracted the first two components of SVD.

### Gene set enrichment analysis of differentially expressed genes under HFD

To seek which biological processes and/or pathways are differentially expressed under a particular dietary intervention, we performed gene set enrichment analysis. The target GSs were obtained from the Molecular Signatures Database (MSigDB) (version 7.4) (Subramanian et al, 2005) using the R package msigdbr (version 7.4.1). Among the datasets in MSigDB, we used Gene Ontology for Biological Process, Hallmark gene sets, KEGG pathways, Reactome pathways, and WikiPathways. Using the sample batch-corrected Gene Score Matrix, we compared nuclei from a pair of dietary conditions, for example, hepatocytes from 4 wk of HFD versus the normal diet. The statistical significance of gene set enrichment was tested using the R package PADOG (version 1.34.0) (Tarca et al, 2012) and by performing permutations for 10,000 times or 20 times the number of GSs in a dataset, whichever was larger. Within each dataset, the GSs that attained a false discovery rate <0.05 were considered statistically significant. Significant GSs were visually inspected using the EnrichmentMap app (version 3.3.3) of the Cytoscape software (version 3.8.2).

### Discovering major modules of TF regulation and assigning biological processes

To elucidate a global picture of TF regulation, we first inferred the regulatory impacts of all TFs on all genes and then extracted major

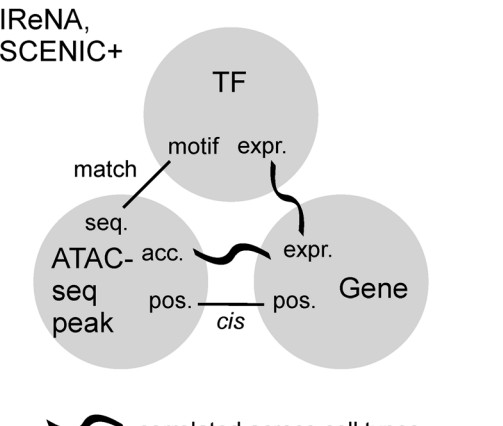

IReNA,
SCENIC+

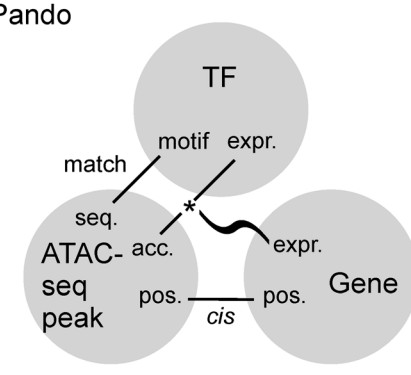

Pando

**Figure 9.  Criteria for defining GRNs are compared across studies.**
In all algorithms, the link between TFs and genes is established by incorporating the information from ATAC-seq peaks. The common requirement is matching the TF's motif with the DNA sequence at an ATAC-seq peak.

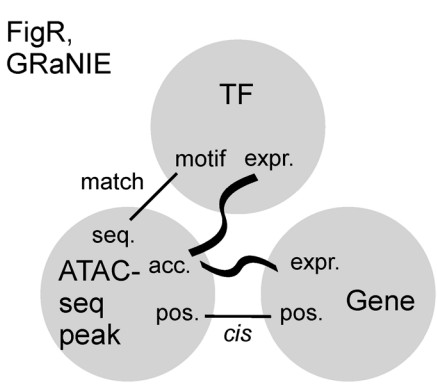

FigR,
GRaNIE

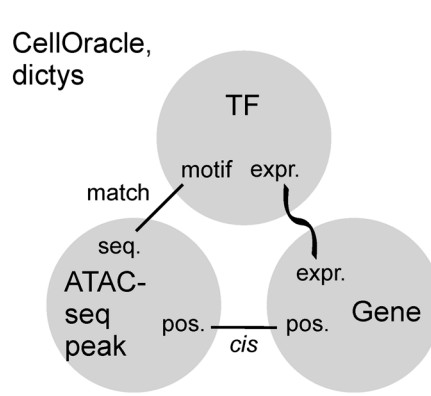

CellOracle,
dictys

This article

acc. = accessibility

expr. = expression

pos. = position

modules of the regulatory relationships between TFs and genes. For the GRN identification in the first step, the relationships between TFs and genes were uncovered by incorporating a third piece of information, chromatin accessibility observed in snATAC-seq, as in previous algorithms (Fleck et al, 2022; González-Blas et al, 2022 *Preprint*; Jiang et al, 2022; Kamal et al, 2022 *Preprint*; Kartha et al, 2022; Wang et al, 2022 *Preprint*; Kamimoto et al, 2023) (Fig 9). Compared with other GRN identification algorithms, our TF-gene

linking criterion was relatively relaxed and yielded more biologically relevant modules (Fig S4), although the sensitivity and specificity have not been validated. For each gene, we computed its regulatory TFs using the R package GENIE3 (version 1.14.0) (Huynh-Thu et al, 2010), which implements a random forest algorithm, thus enabling modeling of nonlinear regulation. The algorithm aims to predict the expression level of the gene based on the binding of TFs and dietary conditions. As for the number of candidate regulators randomly selected at each node of the random forests, we chose the square root of the number of predictors, as suggested in the GENIE3 publication. GENIE3 quantified the impacts of regulators by the proportion of explained variance. By combining the results from all genes, we obtained a regulator–regulatee matrix, in which the rows correspond to TFs and dietary conditions, the columns correspond to genes, and an element represents the importance of a TF to a gene.

Next, we applied nonnegative matrix factorization implemented in the R package NMF (version 0.23.0) (Gaujoux & Seoighe, 2010) to the regulator–regulatee matrix. We used the Brunet algorithm (Brunet et al, 2004) with 100 random starts. The number of factors was chosen based on the data as the inflection point of residual error (Hutchins et al, 2008). In the output of NMF, the membership of TFs (regulators) and genes (regulatees) to a factor was defined by nonnegative weight. We termed these factors as "modules" of TF regulation.

We discovered biological processes that characterize a module by applying gene set enrichment analysis to the genes belonging to the module. For each TF regulation module, we tested if heavily weighted genes were enriched in certain GSs. For each GS (see the above section), we computed the $P$-value and $t$-statistics to test if genes in the given GS had a larger weight than other genes. To account for multiple testing by the GSs, we performed 100 trials of permuting gene names and computed family-wise error rates. The GSs that attained a family-wise error rate of <0.05 were considered statistically significant. The permutation test does not require the assumption of normality.

### GS activity score in single nuclei

To infer the activity of a GS in single nuclei, we computed the "average" score of TF binding and gene expression, with regard to the TFs/genes included in the GS. Formally, we defined the weight of TFs/genes as one if it was included in the GS and as zero otherwise. We then performed SVD for the TF binding and gene expression matrix incorporating the weight. In the Motif Matrix (see above), each TF (row) was standardized by subtracting the mean and being divided by the SD and then multiplied by the weight of the TF. Similarly, in the Gene Score Matrix (see above), each gene (row) was standardized by subtracting the mean and being divided by the SD and then multiplied by the weight of the gene. By stacking the two matrices, we obtained a large matrix, in which the rows correspond to TFs and genes and the columns correspond to the nuclei. Finally, SVD was applied and the first right-singular vector was defined as the GS activity score in single nuclei.

Although the abovementioned activity score can always be computed, it would be meaningless if the TFs/genes in the GS were not co-expressed. Therefore, we conducted a statistical test to exclude such meaningless GS. We checked whether the TFs/genes in the GS were mutually correlated more than by chance. We computed the null distribution of the first singular value by performing 100 trials of the SVD procedure, but with the weight of TFs permutated and the weight of genes permutated and verified whether the actual first singular value could attain an empirical $P$-value of <0.05. The permutation test does not require the assumption of normality.

The principal component approach was adapted from the PLAGE program (Tomfohr et al, 2005) that was developed to compute the pathway activity score of a GS in bulk-tissue specimens. Instead, we applied it to nuclei. Compared with the number of specimens in a bulk experiment, the number of nuclei is much larger, which enabled significance testing by permutation.

## Data Availability

The single-nucleus and bulk ATAC-seq raw data from this publication have been deposited to the DDBJ database [https://www.ddbj.nig.ac.jp] and assigned the identifiers DRA014511 and DRA014458 within BioProject PRJDB13870. Datasets after cleaning have been deposited to figshare [https://figshare.com] and assigned the http://doi.org/10.6084/m9.figshare.20236509. The code generated during this study is available at GitHub https://github.com/fumi-github/rat_singlecell_liver_ArchR.

## Supplementary Information

## Acknowledgements

The authors thank the Center for Omics and Bioinformatics, Graduate School of Frontier Sciences, University of Tokyo and Azenta for their assistance with snATAC-seq. The authors thank Enago (www.enago.jp) for providing an English language review. This work was supported by the NCGM Intramural Research Fund (20A1013, 21A02, 22A1003) and by JSPS KAKENHI [Grant number JP16K07218]. The funding body had no role in the design of the study and collection, analysis and interpretation of data, and in writing the article.

### Author Contributions

F Takeuchi: conceptualization, software, formal analysis, investigation, and writing—original draft, review, and editing.

Y-Q Liang: conceptualization, investigation, and writing—review and editing.

H Shimizu-Furusawa: conceptualization, investigation, and writing—review and editing.

M Isono: investigation and writing—review and editing.

MY Ang: investigation and writing—review and editing.

K Mori: writing—review and editing.

T Mori: writing—review and editing.

E Kakazu: investigation and writing—review and editing.

S Yoshio: writing—review and editing.

N Kato: conceptualization, supervision, and writing—review and editing.

## Conflict of Interest Statement

The authors declare that they have no conflict of interest.

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
