## [Reviewer comments · Life Science Alliance]

Life Science Alliance

Gene regulation modules in non-alcoholic fatty liver disease revealed by single-nucleus ATAC-seq

Fumihiko Takeuchi, Yi-Qiang Liang, Hana Shimizu-Furusawa, Masato Isono, Mia Ang, Kotaro Mori, Taizo Mori, Eiji Kakazu, Sachiyo Yoshio, and Norihiro Kato

DOI: <https://doi.org/10.26508/lsa.202301988>

Corresponding author(s): Fumihiko Takeuchi, National Center For Global Health and Medicine

Review Timeline:	Submission Date:	2023-02-13
	Editorial Decision:	2023-03-30
	Revision Received:	2023-06-03
	Editorial Decision:	2023-07-10
	Revision Received:	2023-07-14
	Accepted:	2023-07-14

Scientific Editor: Novella Guidi

Transaction Report:

March 30, 2023

Re: Life Science Alliance manuscript #LSA-2023-01988-T

Fumihiko Takeuchi
National Center for Global Health and Medicine

Dear Dr. Takeuchi,

Thank you for submitting your manuscript entitled "Single-nucleus ATAC-seq elucidates major modules of gene regulation in the development of non-alcoholic fatty liver disease" to Life Science Alliance. The manuscript was assessed by expert reviewers, whose comments are appended to this letter. We invite you to submit a revised manuscript addressing all the Reviewer comments.

Thank you for this interesting contribution to Life Science Alliance. We are looking forward to receiving your revised manuscript.

Sincerely,

B. MANUSCRIPT ORGANIZATION AND FORMATTING:

Reviewer #1 (Comments to the Authors (Required)):

1. Short summary

This manuscript by Takeuchi et al dissects the changes in liver cellular communities that occur in an animal model of non-alcoholic fatty liver disease (NAFLD). Cellular identities were assessed based on single-nucleus and bulk ATAC sequencing. The study has an elegant design combining matched snATAC-seq and bulk ATAC-seq across conditions. The dataset presented in this manuscript could be of interest to researchers working in genomics of liver disease. However, there are important misinterpretations in the way how ATAC-seq data is presented and discussed that make the manuscript not suitable for publication in its current form. Furthermore, there is no clear advancement in the understanding of NAFLD pathology, as the authors themselves point out. Lastly, the novelty of the new analytical approaches is not demonstrated by benchmarking against already existing methods.

2. Main points

There is a major misinterpretation of the meaning of ATAC-seq data throughout the main text and figure annotations:

In line 62, the weak points of ChIP-seq are mentioned as an argument to perform snATAC-seq. The authors should become aware that there are currently several methods that enable the detection of TF binding sites at the single-cell level.

In line 67 and throughout the manuscript the authors should correct the text to reflect that chromatin accessibility does neither equal TF binding nor gene expression. From ATAC-seq data, we can derive a set of genomic regions or peaks in which we can predict the binding of TFs by TF motif enrichment analysis. This should not be mistaken with detecting the actual binding of TFs. The same is true for gene expression. We cannot directly derive gene expression from ATAC-seq data. All we can do is to estimate gene expression from accessibility at gene body and associated regulatory regions, e.g. promoters. The authors are using gene scores computed with ArchR and this should be clear when reading the main text.

Basic quality control metrics for ATAC-seq data such as number of peaks or fraction of reads in peaks (FRiP) are not reported and should be included.

Methods to identify gene regulatory networks and regulons are available and it seems like the authors do not provide a comparison of their approach compared to other existing methods e.g. <https://morris-lab.github.io/CellOracle.documentation/>; <https://github.com/pinellolab/dictys>; <https://scenicplus.readthedocs.io/en/latest/>). The GENIE3 method used in the paper identifies groups of genes that are coexpressed with TFs, however, coexpression may reflect indirect effects and false positives (<https://www.nature.com/articles/nmeth.4463>). The authors should consider using the TF motif enrichment data as an orthogonal line of evidence to robustly detect regulons.

3. Additional issues

Please, check if the ethics committee that provided approval for the study should be mentioned.

In figure 1E), please indicate that these proportions are based on deconvoluted bulk samples and state the statistical test used and the cut-off used for determining the significance shown with asterisks in the figure.

Figure 3 to 6 are missing adequate annotation of the "gene module" heatmaps as well as appropriate figure legends.

Reviewer #2 (Comments to the Authors (Required)):

The manuscript is well written and found satisfactory with the scope to determine the gene regulation pathways in non-alcoholic fatty liver disease.

We thank the reviewers for his/her time and constructive comments, which we have incorporated into the revised manuscript.

Response to reviewer #1

1. Short summary

This manuscript by Takeuchi et al dissects the changes in liver cellular communities that occur in an animal model of non-alcoholic fatty liver disease (NAFLD). Cellular identities were assessed based on single-nucleus and bulk ATAC sequencing. The study has an elegant design combining matched snATAC-seq and bulk ATAC-seq across conditions. The dataset presented in this manuscript could be of interest to researchers working in genomics of liver disease. However, there are important misinterpretations in the way how ATAC-seq data is presented and discussed that make the manuscript not suitable for publication in its current form. Furthermore, there is no clear advancement in the understanding of NAFLD pathology, as the authors themselves point out. Lastly, the novelty of the new analytical approaches is not demonstrated by benchmarking against already existing methods.

2. Main points

There is a major misinterpretation of the meaning of ATAC-seq data throughout the main text and figure annotations:

In line 62, the weak points of ChIP-seq are mentioned as an argument to perform snATAC-seq. The authors should become aware that there are currently several methods that enable the detection of TF binding sites at the single-cell level.

We corrected the description.

Introduction (Line 61): ChIP-seq/Cut&Run/Cut&Tag assays can be performed on single cells but they can only measure the binding of one type of TF at a time (Salma et al, 2023).

In line 67 and throughout the manuscript the authors should correct the text to reflect that chromatin accessibility does neither equal TF binding nor gene expression. From ATAC-seq data, we can derive a set of genomic regions or peaks in which we can predict the binding of TFs by TF motif enrichment analysis. This should not be mistaken with detecting the actual binding of TFs. The same is true for gene expression. We cannot directly derive gene expression from ATAC-seq data. All we can do is to estimate gene expression from accessibility at gene body and associated regulatory regions, e.g. promoters. The authors are using gene scores computed with ArchR and this should be clear when reading the main text.

We corrected the description throughout the manuscript, stating “binding score” and “expression score”. In addition, we added explanations in the Results section.

Results (Line 148): The expression level of a gene in a nucleus was quantified as a score based on chromatin opening in the gene body and its vicinity.

Results (Line 175): The genome-wide binding of a TF in a nucleus was quantified as a score based on the enrichment of the TF’s motif in the open chromatin.

Basic quality control metrics for ATAC-seq data such as number of peaks or fraction of reads in peaks (FRiP) are not reported and should be included.

We added the information.

Methods (Line 609): We identified 599,035 chromatin accessibility peaks with a width of 501 bp. The fraction of reads in peaks (FRiP) was 0.444 (sample m154207), 0.407 (m154211), 0.541 (m167108), 0.446 (m167203) and 0.557 (m168101). The proportion of reads that aligned within 50bp of TSS was 0.0286 (sample m154207), 0.0224 (m154211), 0.0604 (m167108), 0.0340 (m167203) and 0.0646 (m168101).

Methods to identify gene regulatory networks and regulons are available and it seems like the authors do not provide a comparison of their approach compared to other existing methods e.g. <https://morris-lab.github.io/CellOracle.documentation/>; <https://github.com/pinellolab/dictys>; <https://scenicplus.readthedocs.io/en/latest/>.

We added comparison in Fig. S4. Additionally, we illustrated the difference in the definitions of GRN in Fig. 9. For the purpose of this study—extract major modules of TF-gene regulation—our definition of GRN appeared to be suitable. We could not run Dictys on our dataset using the computers available to us.

Fig. 9. and Fig. S4 are shown in the following pages.

Figure 9: Criteria for defining GRNs are compared across studies
 In all algorithms, the link between TFs and genes is established by incorporating the information from ATAC-seq peaks. The common requirement is matching the TF's motif with the DNA sequence at an ATAC-seq peak.

Figure S4

The effect of GRN algorithms on TF module discovery

As GRN identification algorithms vary in their definitions of GRN (Figure 9), we compared SCENIC+ and CellOracle with our algorithm. Our algorithm's TF-gene linking criterion is relatively relaxed, while SCENIC+ applies a more stringent criterion. Our algorithm requires correlation between ATAC-seq peak accessibility and gene expression, which CellOracle does not require. First, the GRN of hepatocytes was computed using each respective algorithm. Then, the same module discovery method was applied to each GRN: four major components were extracted via nonnegative matrix factorization, and relevant biological processes were detected by gene set enrichment analysis. In SCENIC+, module 3 displayed a strong signal for steroid metabolism, while the other modules presented modest signals for cell adherence, innate immunity and inflammation. CellOracle did not reveal any statistically significant biological process. Our algorithm identified three modules with strong signals for steroid metabolism, inflammation, and lobular zonation.

The GENIE3 method used in the paper identifies groups of genes that are coexpressed with TFs, however, coexpression may reflect indirect effects and false positives (<https://www.nature.com/articles/nmeth.4463>). The authors should consider using the TF motif enrichment data as an orthogonal line of evidence to robustly detect regulons.

We used chromVAR to quantify the TF motif enrichment. We added description.

Methods (Line 825): For each TF, the enrichment of its motif in the peaks was quantified by the chromVAR (Schep et al, 2017) deviation z-score in ArchR.

3. Additional issues

Please, check if the ethics committee that provided approval for the study should be mentioned.

The editorial policy of LSA states "The manuscript must include a statement in the Materials and Methods identifying the institutional and/or licensing committee approving the experiments."

Methods (Line 499): The animal experiments and the protocol were registered and approved by the Animal Research Committee of NCGM (permission number 21009).

In figure 1E), please indicate that these proportions are based on deconvoluted bulk samples and state the statistical test used and the cut-off used for determining the significance shown with asterisks in the figure.

Legend for Figure 1E (Line 976): The proportion in bulk tissue ATAC-seq samples was imputed by deconvolution (n=4 samples per condition). For each cell type, the difference in proportion among dietary conditions was tested by a linear regression where the dependent variable was the imputed composition and the independent variables were indicators of dietary condition. Asterisks (*) indicates statistically significant changes from normal diet under the significance level of $0.05/[7 \times 3]$, adjusted for multiple testing.

Figure 3 to 6 are missing adequate annotation of the "gene module" heatmaps as well as appropriate figure legends.

Legend for Figure 3A (Line 995): The central heatmap visualizes the regulatory impact of TF binding (columns) on genes (rows). While the impact was computed between all TFs and all genes, only representative TFs and genes are plotted for visibility. Subgroups of TFs/genes demonstrating strong within-group regulation were extracted as modules, which are highlighted by red rectangles. The heatmap at the bottom shows the membership of TFs

(columns) to modules (rows). The heatmap on the right displays the membership of genes (rows) to modules (columns). The heatmaps only include representative TFs/genes with membership rankings in the top five and membership weights exceeding half of the top.

Response to Reviewer #2

The manuscript is well written and found satisfactory with the scope to determine the gene regulation pathways in non-alcoholic fatty liver disease.

Thank you for acknowledging the quality of the manuscript.

July 10, 2023

RE: Life Science Alliance Manuscript #LSA-2023-01988-TR

Dr. Fumihiko Takeuchi
National Center For Global Health and Medicine
Research Institute
1-21-1 Toyama
Shinjuku-ku, Tokyo 162-8655
Japan

Dear Dr. Takeuchi,

Thank you for submitting your revised manuscript entitled "Gene regulation modules in non-alcoholic fatty liver disease revealed by single-nucleus ATAC-seq". We would be happy to publish your paper in Life Science Alliance pending final revisions necessary to meet our formatting guidelines.

- please address the final Reviewer 1's points
- please upload your Tables in editable .doc or excel format;
- please add the Twitter handle of your host institute/organization as well as your own or/and one of the authors in our system
- please add your main, supplementary figure, and table legends to the main manuscript text after the references section;
- all figure legends should only appear in the main manuscript file
- we encourage you to revise the figure legends for figures 4, 5, and 6 such that the figure panels are introduced in alphabetical order;
- please add callouts for Figure S3A and B to your main manuscript text

A. FINAL FILES:

B. MANUSCRIPT ORGANIZATION AND FORMATTING:

Sincerely,

Reviewer #1 (Comments to the Authors (Required)):

Response to authors from reviewer #1 (round 2)

The authors have addressed most of the comments and with final editorial work the manuscript should be ready for publication.

Main points

The previously described major 5 points are summarized below.

1) Acknowledging the existence of single-cell TF binding detection methods.

The authors have now modified the text to reflect this and added a citation. It is still not correct to say that these methods "can only measure the binding of one type of TF at a time" since, as indicated in the review they cite, some of these methods can detect multiple TF binding sites simultaneously at the single-cell level. Still, it is true that sc/snATAC-seq is one of the most comprehensive methods to predict TF activity.

2) TF binding and gene expression are not directly measured by scATAC-seq.

The authors have modified the text accordingly. Some figure annotations still show the "TF binding" label and should be modified.

3) QC metrics.

This comment has been satisfactorily addressed.

4) GRN method comparison.

The addition of Figure 9 may not be necessary, especially as not all methods shown were applied to the data. However, the comparison shown in Figure S4 is valuable. The authors added the following statement in the main text about Figure S4:

"Compared to other GRN identification algorithms, our TF-gene linking criterion was relatively relaxed, yielding more biologically relevant modules (Figure S4)."

This statement is not supported by any validation and should be rephrased to account for the possibility that their method is more sensitive but potentially also less specific.

5) GENIE3 co-expression.

This comment has been satisfactorily addressed.

Additional points

The annotation of heatmaps in figures 3 to 6 still needs to be completed, e.g. annotating the color bar

July 14, 2023

RE: Life Science Alliance Manuscript #LSA-2023-01988-TRR

Dr. Fumihiko Takeuchi
National Center For Global Health and Medicine
Research Institute
1-21-1 Toyama
Shinjuku-ku, Tokyo 162-8655
Japan

Dear Dr. Takeuchi,

Thank you for submitting your Research Article entitled "Gene regulation modules in non-alcoholic fatty liver disease revealed by single-nucleus ATAC-seq". It is a pleasure to let you know that your manuscript is now accepted for publication in Life Science Alliance. Congratulations on this interesting work.

DISTRIBUTION OF MATERIALS:

Again, congratulations on a very nice paper. I hope you found the review process to be constructive and are pleased with how the manuscript was handled editorially. We look forward to future exciting submissions from your lab.

Sincerely,
